# A Generative Diffusion Framework for Single Image Reflection Separation

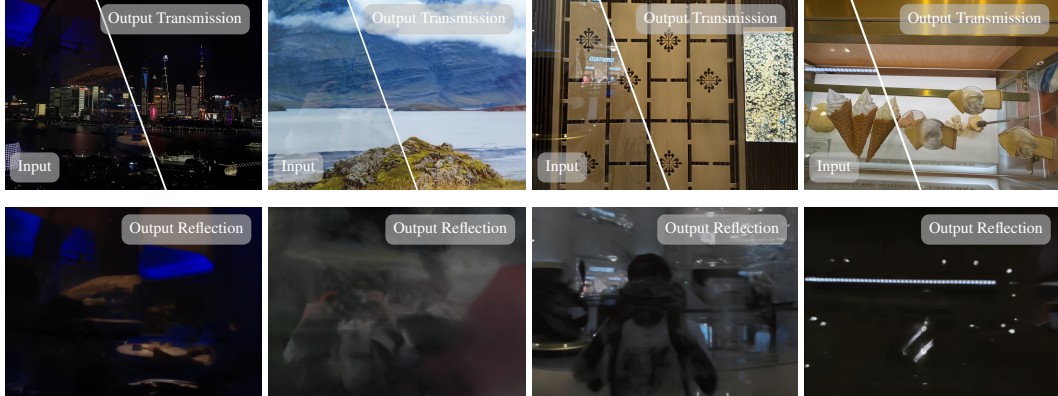

Figure 1: **Given an input image containing reflections, our method separates it into distinct transmission and reflection layers.** By effectively leveraging generative diffusion priors, our method delivers strong and reliable separation performance even in challenging scenarios, including scenes with strong reflections (*first* and *fourth* columns) by hallucinating missing details, as well as scenes with subtle reflections, accurately extracting meaningful reflection information.

## Abstract

Single-image reflection separation remains challenging due to its ill-posed nature, especially under extreme conditions with strong or subtle reflections. Existing methods often struggle to recover both layers in glare or weak-reflection scenarios, because of insufficient information. This paper presents the first diffusion model explicitly fine-tuned for this task, leveraging generative diffusion priors for robust separation. Our method simultaneously generates transmission and reflection layers through a unified diffusion model, incorporating a novel cross-layer self-attention mechanism for better feature disentanglement. We further introduce a disjoint sampling strategy to iteratively reduce interference between the layers during diffusion and a latent optimization step with a learned composition function for improved results in complex real-world scenarios. Extensive experiments show our approach achieves superior separation performance on multiple real-world benchmarks and surpasses state-of-the-art methods in both quantitative metrics and perceptual quality.

## 1 Introduction

Images captured through semi-reflective media often exhibit a complex superposition of transmission (the intended scene) and reflection (the undesired or secondary scene) layers. This layered mixture not only diminishes visual clarity but can also severely degrade the performance of downstream computer vision tasks by obscuring important image content or introducing misleading visual cues that can confound computational analysis (Chang et al., 2020a; Qiu et al., 2023; Wan et al., 2021). Despite extensive research, single-image reflection separation remains a significant and challenging problem in computer vision due to its inherently ill-posed nature.

To mitigate this ill-posedness, numerous approaches have incorporated additional cues, such as flash illumination (Lei & Chen, 2021; Chang et al., 2020b), extra frames (Xue et al., 2015; Liu et al.,

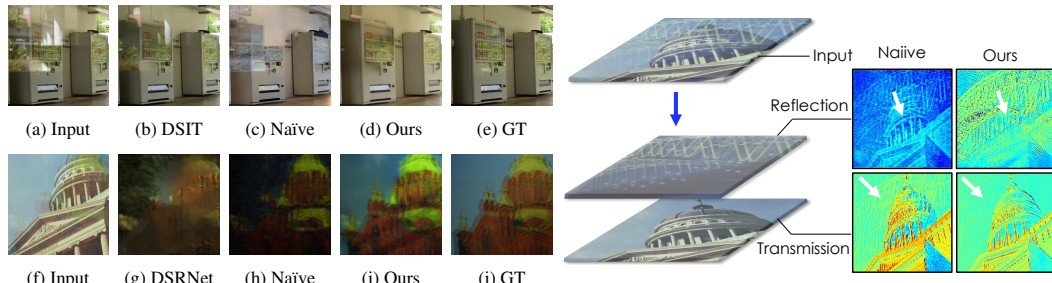

Figure 2: **Motivation.** Existing state-of-the-art methods, such as (b) DSIT (Hu et al., 2025) and (g) DSRNet (Hu & Guo, 2023b), struggle with challenging scenarios like strong reflections or complex overlapping content, resulting in residual reflections or distorted outputs. (c, h) Naïvely adapting diffusion models for single-layer prediction can hallucinate some missing information but often introduces unrealistic artifacts and inaccuracies. (d, i) In contrast, our method leverages generative priors to jointly model transmission and reflection layers, significantly improving focus on relevant features (*right*).

2020; Chugunov et al., 2024), and language (Zhong et al., 2024; Hong et al., 2024b) to address the issue. However, the requirement for additional data limits their practical applicability. Meanwhile, deep learning-based single-image reflection separation methods (Hu & Guo, 2023b; Hu et al., 2025) struggle to recover images in glare and to separate reflections when the reflection signal is weak.

Motivated by the limitations observed in previous approaches (see Figure 2), we propose the first approach that explicitly fine-tunes diffusion models for the reflection separation task. Our method leverages powerful generative priors from the pre-trained diffusion model, effectively addressing challenging cases encountered by previous methods. In high-reflection scenarios, our model adeptly hallucinates missing details, while in subtle or low-light reflection scenarios, it accurately captures meaningful reflection content (see Figure 1).

We extract noise predictions explicitly associated with transmission and reflection layers, enabling effective reflection reconstruction. To further enhance separation, we introduce a diffusion sampling strategy where predicted noises from each layer mutually constrain each other, reducing joint artifacts and facilitating clear layer disentanglement. Furthermore, to improve performance in challenging "in-the-wild" scenarios, we integrate latent optimization with a composition loss in the latent space. This strategy substantially reduces computational overhead and GPU memory requirements by optimizing a composition function in the latent space while simultaneously boosting separation quality by preserving information and improving layer disentanglement. Our key contributions include:

- Introducing the first diffusion-model-based reflection separation method simultaneously fine-tuned for transmission and reflection layers, effectively exploiting generative priors.
- Proposing cross-layer self-attention and disjoint sampling to enhance the diffusion model's reflection separation capability by improving information interaction across layers and optimizing the sampling strategy.
- Incorporating latent optimization with a composition loss, enabling robust reflection separation under challenging conditions while greatly reducing computational resources.

## 2   RELATED WORK

**Diffusion Models for Image Processing.**   Diffusion models (Ho et al., 2020; Rombach et al., 2022; Song et al., 2020; Nichol et al., 2021) have shown remarkable capabilities in generating high-quality images, with growing interest in adaptation for image processing tasks. Recent theoretical work (Park et al., 2024) unifies various diffusion formulations through Tweedie's formula, while comprehensive surveys (Li et al., 2025) categorize their applications in restoration tasks. Current approaches fall into two main categories.

On the one hand, *training-based* methods adapt pre-trained diffusion models by fine-tuning them to accept image conditions. Key implementations include ControlNet (Zhang et al., 2023), which

introduces condition modules for flexible control, with recent advances like ControlNet++ (Li et al., 2024) improving conditional consistency through efficient feedback mechanisms and InnerControl (Konovalova et al., 2025) enforcing spatial consistency across all diffusion steps. Direct input concatenation methods (Ke et al., 2024; Wang et al., 2024c; Zeng et al., 2024) and high-fidelity guided synthesis approaches (Singh et al., 2023) have also shown success. Recent work on cross-attention mechanisms (Yang et al., 2024; Liu et al., 2024; Shentu et al., 2024) demonstrates their effectiveness for feature disentanglement, which motivates our cross-layer attention design.

On the other hand, *training-free* methods leverage inherent model priors during inference without additional training. Beyond early approaches (Chung et al., 2022; Wang et al., 2022), recent advances include sophisticated inverse problem solvers using second-order Tweedie approximations (Rout et al., 2024), optimal control formulations (Li & Pereira, 2024), plug-in methods robust to unknown noise (Wang et al., 2024a), and moment-projected diffusions with theoretical guarantees (Boys et al., 2023). Test-time optimization has evolved through SMC-based alignment (Kim et al., 2025), multi-objective Pareto-guided generation (Yao et al., 2024), and noise trajectory search (Ma et al., 2025). However, these methods still rely on accurately modeling the imaging formulation and struggle with complex scenarios like reflection separation.

Our approach combines both strategies: fine-tuning for better condition integration and initial stable results, followed by test-time optimization to refine outputs using the model's implicit knowledge. This hybrid strategy, supported by recent latent-space optimization techniques (Hong et al., 2024a), reduces sensitivity to initial values while improving convergence stability and result quality.

**Single-image Reflection Separation.** Single-image reflection separation extracts transmission and reflection layers from mixed images challenged by signal overlap and complex lighting conditions. Recent surveys (Yang et al., 2025) identify data scarcity and generative model integration as key remaining challenges. Methods for reflection removal can be broadly classified into three categories:

(i) *Traditional methods.* Early approaches relied on physical models or hand-crafted priors (e.g., edge sparsity, relative smoothness) to constrain the problem (Levin et al., 2002; 2004; Levin & Weiss, 2007; Li & Brown, 2014; Yang et al., 2019). Although intuitive, these assumptions restrict applicability and often lead to unstable results in complex scenarios, struggling with both global structures and local details.

(ii) *Deep learning methods.* Recent advances harness large-scale datasets to learn the underlying distributions of transmission and reflection layers (Dong et al., 2021; Fan et al., 2017; Lei et al., 2020; Li et al., 2020; 2023; Wei et al., 2019; Yang et al., 2018; Zhang et al., 2018; Hu & Guo, 2021; Song et al., 2023; Zhu et al., 2024; Hu et al., 2025; Hu & Guo, 2023b; Zhong et al., 2024; Wan et al., 2019; Feng et al., 2021). CEILNet (Fan et al., 2017) introduced edge detection with relative smoothness assumptions, while Zhang et al. (2018) and ERRNet (Wei et al., 2019) used perceptual losses and non-aligned training data. BDN (Yang et al., 2018), CoRRN (Wan et al., 2019), and the recent RRW (Zhu et al., 2024) with its Maximum Reflection Filter (MaxRF) explored various architectures and loss functions. Dual-stream interactive architectures have further enhanced separation quality: IBCLN (Li et al., 2020) employed convolutional LSTM networks, YTMT (Hu & Guo, 2021) introduced interactive feature exchange, and DSRNet (Hu & Guo, 2023b) advanced the approach with an efficient MuGI module. Recent methods have introduced learnable residue terms for non-linear formulations (Hu & Guo, 2023a), language-guided separation with cross-attention (Zhong et al., 2024), and neural spline fields for multi-frame scenarios (Chugunov et al., 2024). Additionally, methods such as RAGNet (Li et al., 2023), DMGN (Feng et al., 2021), RobustSIRR (Song et al., 2023), DSIT (Hu et al., 2025), and RDNet (Zhao et al., 2025) have tackled specific challenges via specialized designs. Despite these advances, deep learning approaches still struggle significantly in challenging scenes with strong or complex reflections, as evidenced by recent benchmarks (Zhu et al., 2024).

(iii) *Generative methods.* Recently, some approaches have employed generative frameworks for reflection removal. Beyond initial attempts (Rosh et al., 2023; Wang et al., 2024b), recent work explores multi-layer decomposition (Tudosiu et al., 2024) and addresses noisy data through consistent diffusion formulations (Daras et al., 2024). However, most of these works only focus on recovering the transmission layer. In contrast, we find that generative priors are also very helpful for extracting meaningful reflection information, not only providing additional scene context but also further enhancing the recovery of the transmission layer.

Figure 3: **Overview of our framework.** Given an input image, we first encode it into a latent representation and then leverage a fine-tuned diffusion model guided by distinct prompts ("*Transmission*" and "*Reflection*") to separate the transmission and reflection layers. We introduce a cross-layer self-attention mechanism that allows effective interaction between the two layers. Our novel disjoint sampling strategy further reduces layer interference by guiding the diffusion trajectory. Finally, latent optimization via a learned latent composition function and composition loss refines the latent representations, significantly improving separation fidelity while maintaining computational efficiency.

The most relevant paper to our work is L-DiffER (Hong et al., 2024b), which fine-tunes a Stable Diffusion model and leverages textual descriptions of each layer to obtain *only a clean transmission*. In contrast, our approach jointly separates the *transmission and reflection layers*. Furthermore, L-DiffER relies on precise and comprehensive language annotations, which are not always available or sufficiently accurate. This reliance may lead to partial information loss and increased annotation costs. Our method instead uses simple fixed prompts ("Transmission" and "Reflection") combined with cross-layer attention and disjoint sampling to achieve superior separation without complex linguistic guidance.

## 3 METHOD

Given an input image $\mathcal{I}$, reflection separation aims to decompose it into a transmission layer ($\mathcal{T}$) and a reflection layer ($\mathcal{R}$). Figure 3 illustrates the proposed framework for this task.

### 3.1 PRELIMINARIES

Our method leverages the pre-trained Latent Diffusion Model (LDM). Therefore, we briefly introduce its internal structure.

**Latent Representation.** LDM uses a pre-trained VAE encoder $\mathcal{E}(\cdot)$ to compress the data into a compact latent: $z_0 = \mathcal{E}(I)$ for further processing. The decoder of the pre-trained VAE converts the latent into pixels $\mathcal{D}(z_0)$.

**Diffusion Process.** A diffusion process gradually corrupts $z_0$ by injecting Gaussian noises over $T$ steps: $z_t = \sqrt{1-\beta_t} z_{t-1} + \sqrt{\beta_t} \epsilon_{t-1} = \sqrt{\alpha_t} z_0 + \sqrt{1-\alpha_t} \epsilon$, resulting in sequential latents $\{z_t\}_{t=1}^T$, with latent $z_T$ is fully noised. $\epsilon, \epsilon_{t-1} \sim \mathcal{N}(0,1)$ and $\alpha_t := \prod_{s=1}^t (1-\beta_s)$.

**Denoising Sampling.** The diffusion model then learns to invert this noising process through a noise-prediction network $\epsilon_\theta(\cdot)$, trained by minimizing $\mathbb{E}_{\epsilon \sim \mathcal{N}(0,1), t, z_0} \left\| \epsilon_t - \epsilon_\theta(z_t, t, c) \right\|_2^2$, where $c$ denotes the text condition used for text-to-image generation. After training, one can iteratively denoise $z_T$ back to a clean latent code through numerical solvers (Song et al., 2020; Ho et al., 2020). To avoid the tedious sampling process, Tweedie's formula enables one-step approximation: $\mathbb{E}[z_0|z_t] := z_{0|t} = (z_t - \sqrt{1-\alpha_t}\epsilon_\theta) / \sqrt{\alpha_t}$.

**Attention.** The noise predictor in LDM is a modified U-Net (Ronneberger et al., 2015) with self-attention for long-range dependencies and cross-attention for text-based conditioning, enabling efficient denoising and high-quality image synthesis. The attention mechanism follows

$H = \text{softmax}(Q\,K/\sqrt{d})\,V$, where $Q$, $K$ and $V$ are projected feature tokens in the U-Net. Specifically, for self-attention, they correspond to spatial feature embeddings, while for cross-attention, $K$ and $V$ are text tokens.

## 3.2 Feedforward Generative Seperation

**Conditional Fine-tuning.** Given a pre-trained LDM, we can follow prior practices on different tasks (Wang et al., 2024c; Ke et al., 2024) to fine-tune it to predict one of the layers conditioned on a real input image $\mathcal{I}$. Specifically, we take the VAE encoding $z^{\mathcal{I}}$ of the input image and concatenate it with the noisy latent $z_t$ along the channel dimension, then feed them into the U-Net. The model is fine-tuned using a combined loss for the reflection and transmission layers, each following the same objective as section 3.1, where $z_t$ in the original objective is replaced by the concatenated tensor. To harness a single diffusion model for both layers, we introduce textual clues, that is, "Transmission" and "Reflection" prompts, to guide the model to predict the corresponding layer.

While fine-tuning LDM to estimate transmission and reflection layers achieves reasonable results, challenging cases still exhibit mutual artifacts, with reflections contaminating the transmission layer and vice versa. To address this, we introduce two enhancements to refine the feedforward separation: **(i)** interaction-driven self-attention for robust feature disentanglement and **(ii)** disjoint sampling to further suppress residual overlaps.

**Cross-Layer Self-Attention.** We modify the diffusion U-Net's self-attention modules to enable explicit cross-layer interactions. Specifically, each attention block jointly processes transmission and reflection features, allowing queries from one layer to attend to keys from both. As illustrated in Figure 4, queries associated with transmission queries dynamically interact with reflection keys and vice versa. Formally, the new attention mechanism is defined as: $H^i = \text{softmax}\left(\frac{Q^i\left[K^{\mathcal{T}};K^{\mathcal{R}}\right]^{\top}}{\sqrt{d}}\right)\left[V^{\mathcal{T}};V^{\mathcal{R}}\right]$, where $i \in \{\mathcal{T},\mathcal{R}\}$, and $Q^i, K^i, V^i$ are the query, key, and value projections activated by the corresponding prompt. The symbol $[\cdot;\cdot]$ denotes concatenation along the spatial dimension. This cross-layer attention structure encourages queries to leverage information from both layers, effectively disentangling the two layers.

By promoting explicit cross-layer interaction, our method guides each layer to separate content distinctly. When reflection signals dominate certain regions, the transmission branch learns to compensate accordingly, mitigating confusion. Conversely, the reflection branch becomes adept at identifying true reflections versus transmission content. Our experiments show that this approach substantially reduces unintended feature overlaps in intermediate representations and final outputs.

**Disjoint Sampling.** To mitigate residual overlap between transmission and reflection layers, we introduce disjoint sampling, inspired by Classifier-Free Guidance (Ho & Salimans, 2022), to explicitly push their latent representations apart (see Figure 5). Formally, let $\epsilon_t^{\mathcal{T}}$ and $\epsilon_t^{\mathcal{R}}$ denote the predicted noise at step $t$ for the transmission and reflection layers, respectively. For transmission generation, disjoint sampling aims to maximize $\frac{p(z_t|\mathcal{T})}{p(z_t|\mathcal{R})^k}$. Fine-tuning allows $\epsilon_t^{\mathcal{T}}$ and $\epsilon_t^{\mathcal{R}}$ to model $\nabla_z \log p(z_t|\mathcal{T})$ and $\nabla_z \log p(z_t|\mathcal{R})$, respectively, enabling noise updates $\hat{\epsilon}_t^{\mathcal{T}} = \epsilon_t^{\mathcal{T}} + k(\epsilon_t^{\mathcal{T}} - \epsilon_t^{\mathcal{R}})$. The denoised latent is then computed by $z_{t-1}^{\mathcal{T}} = \frac{1}{\sqrt{1-\beta_t}}\left(z_t^{\mathcal{T}} - \frac{\beta_t}{\sqrt{1-\alpha_t}}\hat{\epsilon}_t^{\mathcal{T}}\right)$, with an analogous update applied to $\epsilon_t^{\mathcal{R}}$ to ensure mutual separation. This sampling strategy effectively maximizes the desired probability ratio, and iterating such steps throughout the diffusion process significantly reduces cross-contamination.

However, such updates might cause undesirable color drifts. To counteract this, we propose a *Fidelity-Guided Feature Modulation* (FGFM) module, which references multi-scale features from the original mixture image. FGFM introduces a parameter $w$ to balance separation quality with original image integrity. The modulated decoding feature is computed as: $\hat{y}_{dec} = y_{dec} + w \times f([y_{enc}\,|\,y_{dec}])$, where $y_{enc}$ denotes encoded features from the original mixture image, $y_{dec}$ denotes decoded features from the current diffusion step, $f(\cdot)$ represents a stack of convolutional layers, $[\cdot\,|\,\cdot]$ denotes concatenation along the channel dimension, and $w$ controls the modulation strength. FGFM training utilizes combined pixel-level losses, with details in the supplementary.

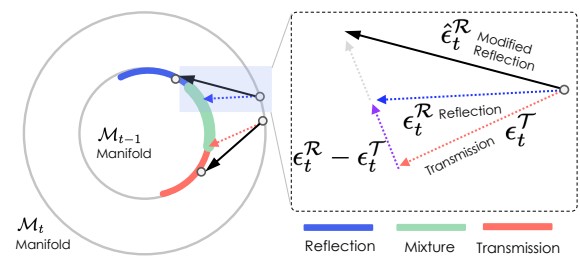

Figure 4: **Cross-layer self-attention mechanism.** Queries from each layer (Transmission or Reflection) attend to keys from both layers, guided by prompts, enabling effective information exchange and clearer separation of transmission and reflection features.

Figure 5: **Disjoint sampling strategy.** Due to latent overlap, naive sampling often causes joint artifacts. Our method explicitly leverages predicted noise differences $\epsilon_t^{\mathcal{T}} - \epsilon_t^{\mathcal{R}}$ as mutual negative guidance, iteratively modifying the diffusion trajectories of transmission and reflection layers, effectively promoting their clear and accurate separation.

### 3.3 SEPARATION OPTIMIZATION BY COMPOSITION

A single forward pass often cannot fully satisfy the composition constraints. We therefore introduce a test-time latent optimization stage to further refine the initially separated layers. Unlike previous methods that assume a pixel-space relationship ($\mathcal{I} = \mathcal{T} + \mathcal{R}$), we learn a latent composition function in the latent space. This formulation more accurately models the composition and achieves better results, while also alleviating non-linear mismatches and significantly reducing both computation time and GPU memory requirements by avoiding backpropagation through the decoder and large feature maps.

**Latent Composition Function.** We train the composition network $\mathcal{C}$ on a synthetic dataset with known ground truth. Each sample contains latents $(z^{\mathcal{T}}, z^{\mathcal{R}})$ of the transmission and reflection layers, plus the latent code $z^{\mathcal{I}}$ of the mixture image. The composition network $\mathcal{C}$ is implemented as a compact convolutional neural network designed for cross-layer interactions: $\mathcal{C}(z^{\mathcal{T}}, z^{\mathcal{R}}) = \mathcal{F}(\mathcal{G}_1(z^{\mathcal{T}}, z^{\mathcal{R}}), \mathcal{G}_2(z^{\mathcal{T}}, z^{\mathcal{R}}))$. We optimize the parameters of the composition network by minimizing $\mathcal{L}_{\text{comp}} = \left\| \hat{z}^{\mathcal{I}} - z^{\mathcal{I}} \right\|_2^2$.

**Test-time Latent Optimization.** At inference time, we obtain the initial latents $(z_t^{\mathcal{T}}, z_t^{\mathcal{R}})$ through our diffusion model. Following section 3.1, we first obtain $(z_{0|t}^{\mathcal{T}}, z_{0|t}^{\mathcal{R}})$ from $(z_t^{\mathcal{T}}, z_t^{\mathcal{R}})$. Subsequently, we generate the pseudo composed latent at time step 0 as $\hat{z}^{\mathcal{I}} = \mathcal{C}(z_{0|t}^{\mathcal{T}}, z_{0|t}^{\mathcal{R}})$. These latents are then iteratively refined to align better with the input. By minimizing $\mathcal{L}_{\text{comp}}$, each update progressively encourages $z^{\mathcal{T}}$ and $z^{\mathcal{R}}$ to reconstruct the mixture in a way consistent with the learned composition model. The optimization is given by $\hat{z}_t^{\mathcal{T}} = z_t^{\mathcal{T}} - \gamma_i ||z_t^{\mathcal{T}}|| \nabla_{z_t^{\mathcal{T}}} \mathcal{L}_{\text{comp}}$, where $\gamma_i$ is the weight to control the update. We derive $\hat{z}_t^{\mathcal{R}}$ using the same formulation. The complete algorithm is provided in the supplementary material.

## 4 EXPERIMENTS

### 4.1 IMPLEMENTATION DETAILS

**Experiment Setting.** We evaluate our method using three real-world datasets, including Real20 (Zhang et al., 2018), Nature20 (Li et al., 2020), and SIR[2] dataset (Wan et al., 2017). Unlike previous works, we also evaluate the reflection results. Since only the SIR[2] dataset includes ground truth for reflections, the quantitative comparison for the reflection layer is conducted solely on this dataset. We select PSNR, SSIM, LPIPS, and DISTS as the evaluation metrics. We use a resolution of $960 \times 960$ during inference with the FGFM parameter $w = 0.8$.

### 4.2 COMPARISON TO STATE-OF-THE-ART METHODS

We compare our method with six recent state-of-the-art approaches: YTMT (Hu & Guo, 2021), RobustSIRR (Song et al., 2023), DSRNet (Hu & Guo, 2023b), RRW (Zhu et al., 2024), DSIT (Hu

Table 1: **Quantitative comparison of the transmission layer across different real-world datasets (Real20 (Zhang et al., 2018), Nature (Li et al., 2020), SIR$^2$ (Wan et al., 2017)).** We compare our method with state-of-the-art approaches. The best and second-best results are highlighted in bold and underline, respectively. Our approach consistently achieves superior performance across most metrics, particularly excelling in perceptual quality measures (LPIPS, DISTS), demonstrating its effectiveness in producing visually cleaner and more accurate transmission layers.

| Dataset (size) | Metric | Method | | | | | | | |
|---|---|---|---|---|---|---|---|---|---|
| | | YTMT | RobustSIRR | DSRNet | RRW | DSIT | RDNet | ControlNet | Ours |
| Real20 (20) | ↑ PSNR | 23.11 | 23.04 | 23.93 | 20.85 | 24.16 | 25.02 | 18.78 | **25.42** |
| | ↑ SSIM | 0.791 | 0.796 | 0.809 | 0.737 | 0.796 | 0.826 | 0.645 | **0.850** |
| | ↓ LPIPS | 0.135 | 0.157 | 0.120 | 0.207 | 0.133 | 0.108 | 0.282 | **0.080** |
| | ↓ DISTS | 0.124 | 0.136 | 0.111 | 0.164 | 0.119 | 0.104 | 0.216 | **0.088** |
| Nature (20) | ↑ PSNR | 24.85 | 25.57 | 26.25 | 25.99 | 26.11 | 26.60 | 20.04 | **26.82** |
| | ↑ SSIM | 0.822 | 0.826 | 0.835 | 0.828 | 0.825 | 0.836 | 0.721 | **0.837** |
| | ↓ LPIPS | 0.091 | 0.122 | 0.102 | 0.120 | 0.121 | 0.083 | 0.209 | **0.052** |
| | ↓ DISTS | 0.086 | 0.106 | 0.095 | 0.113 | 0.106 | 0.080 | 0.168 | **0.066** |
| SIR$^2$ (454) | ↑ PSNR | 24.13 | 23.96 | 24.18 | 23.30 | 25.31 | **25.87** | 20.77 | 25.62 |
| | ↑ SSIM | 0.887 | 0.875 | 0.901 | 0.862 | **0.911** | 0.909 | 0.812 | **0.911** |
| | ↓ LPIPS | 0.087 | 0.126 | 0.074 | 0.121 | 0.069 | 0.075 | 0.144 | **0.049** |
| | ↓ DISTS | 0.088 | 0.109 | 0.078 | 0.107 | 0.078 | 0.075 | 0.138 | **0.067** |

Figure 6: **Qualitative comparison of our method with state-of-the-art approaches on background-reflection separation using real-world images.** Compared to other methods, our approach yields cleaner and more accurate transmission (background) layers, effectively removing reflection artifacts and simultaneously recovering clearer and more meaningful reflection details, demonstrating significant improvements in challenging real-world scenarios. More results can be found in the supplementary material.

et al., 2025), and RDNet (Zhao et al., 2025). Additionally, we include ControlNet (Zhang et al., 2023) as a diffusion-based baseline due to the absence of publicly available implementations for other related diffusion-based methods. For a fair comparison, we retrained all methods using their official training code on the same dataset and evaluated them using identical metrics. Evaluation with the official pre-trained weights is presented in the supplementary material. Notably, our retrained versions outperform their corresponding pre-trained counterparts across multiple metrics.

**Quantitative Comparison.** Table 1 shows that for the transmission component, our approach achieves the best performance across most metrics, particularly in perceptual performance, significantly outperforming previous methods on both the Real20 and Nature20 datasets. Table 2 demonstrates that for the reflection component, our scores are markedly superior to those of earlier approaches.

**Qualitative Comparison.** We also provide visual comparisons of the transmission and reflection layers generated by different methods. Note that RobustSIRR (Song et al., 2023) and RRW (Zhu et al., 2024) are excluded here since they only produce transmission layers. As illustrated in Figure 6, our method recovers meaningful reflection layers, accurately capturing even subtle details from the input image, thereby producing cleaner transmission layers. Additionally, in cases involving strong specular reflections, our approach effectively reconstructs the underlying information in the transmission layer, compensating reasonably for the missing content.

Table 2: **Quantitative comparison of the reflection layer.** Comparison with SOTA on SIR[2] (Wan et al., 2017). Best and second-best are **bold** and underlined.

| Metric | YTMT | DSRNet | DSIT | RDNet | Ours |
|---|---|---|---|---|---|
| ↑PSNR | 16.64 | 20.59 | 18.51 | 18.00 | **21.12** |
| ↑SSIM | 0.252 | 0.671 | 0.462 | 0.362 | **0.681** |
| ↓LPIPS | 0.414 | 0.367 | 0.347 | 0.364 | **0.248** |
| ↓DISTS | 0.578 | 0.387 | 0.399 | 0.337 | **0.276** |

Table 3: **Ablation study.** Contribution of Cross-layer self-Attention (C), Latent Optimization (O), and Disjoint Sampling (D) on the SIR[2] dataset. The complete model (C+O+D) achieves the best metrics.

| C | O | D | PSNR ↑ | SSIM ↑ | LPIPS ↓ | DISTS ↓ |
|---|---|---|---|---|---|---|
| | | | 24.93 | 0.843 | 0.095 | 0.107 |
| ✓ | | | 24.93 | 0.858 | 0.083 | 0.096 |
| ✓ | ✓ | | 25.33 | 0.866 | 0.079 | 0.092 |
| ✓ | ✓ | ✓ | **25.62** | **0.911** | **0.049** | **0.067** |

Input w/o w/ GT

Figure 7: **Ablation study of cross-layer self-attention.** Incorporating cross-layer self-attention promotes inter-layer information interaction, improving reflection (*bottom*) quality and consequently enhancing the perceptual clarity of the transmission layer (*top*).

w/o w/ w/o w/
└── Transmission ──┘ └── Reflection ──┘

Figure 8: **Ablation study of separation sampling.** Without separation sampling, noticeable artifacts appear due to mutual interference between transmission and reflection. Incorporating separation sampling significantly reduces artifacts, improving separation clarity.

## 4.3 ABLATION STUDY

In this section, we validate the contribution of each module, including Cross-Layer Self-Attention, Latent Optimization, and Disjoint Sampling. We also test the sensitivity of the parameter used in FGFM.

**Cross-Layer Self-Attention.** Without cross-layer self-attention, we observe insufficient interaction between reflection and transmission layers. As illustrated in Figure 7, introducing cross-layer self-attention enables comprehensive information exchange between layers, significantly enhancing the quality of reflection reconstruction. An improved reflection layer, in turn, provides stronger guidance to the transmission layer, resulting in clearer outputs. As further

Table 4: **Ablation study on cross-layer self-attention (CLSA).** We evaluate the effectiveness of CLSA in predicting the reflection layer on SIR[2] dataset (Wan et al., 2017).

| | PSNR ↑ | SSIM ↑ | LPIPS ↓ | DISTS ↓ |
|---|---|---|---|---|
| w/o CLSA | 20.81 | **0.688** | 0.295 | 0.381 |
| w/ CLSA | **20.87** | 0.659 | **0.252** | **0.285** |

validated by the perceptual metrics in Table 3, the transmission performance exhibits noticeable improvement. This mechanism also helps recover reflection layers, as shown in Table 4.

**Latent Optimization.** This approach effectively enforces the composite image constraint, ensuring that the generated transmission and reflection layers remain semantically closer to the original image. As illustrated in Figure 9, latent optimization partially restores missing information. From Table 3, we observe that latent optimization consistently improves performance across all metrics. In Table 5, we further compare our latent-space optimization method with pixel-space optimization constrained by $\mathcal{I} = \mathcal{T} + \mathcal{R}$. At an input resolution of $512 \times 512$, each update step takes only 0.15 seconds using our method, whereas pixel-space optimization requires 1.53 seconds. This shows that our approach achieves significantly better performance with reduced runtime and memory consumption.

**Disjoint Sampling.** Figure 8 reveals that, in the original results, the reflection layer inadvertently contains some transmission content (cups and a flower pot). After using separation sampling, the results become more accurate. This improvement is likely because, during inference, the two layers can reference each other's information, gradually adjusting the diffusion direction to reduce mutual interference between layers. The figure shows that, while the original reflection still retained traces of transmission objects, the proposed approach significantly mitigates this issue. Quantitative results

Table 5: **Ablation study on optimization methods.** We compare our latent-space optimization method with pixel-space optimization using $\mathcal{I} = \mathcal{T} + \mathcal{R}$. Our approach achieves superior performance, is significantly faster, and consumes less memory. Experiments are conducted on the Nature (Li et al., 2020) dataset at a lower resolution.

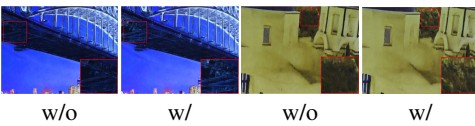

w/o          w/          w/o          w/

Figure 9: **Ablation study of the latent optimization.** Compared to results without latent optimization (w/o), applying latent optimization (w/) significantly improves visual quality by recovering clearer details and reducing artifacts, highlighting its importance in refining the final transmission layer output.

|  | PSNR ↑ | SSIM ↑ | LPIPS ↓ | DISTS ↓ |
|---|---|---|---|---|
| Pixel OP | 21.53 | 0.715 | 0.107 | 0.120 |
| Latent OP | **25.57** | **0.783** | **0.106** | **0.110** |

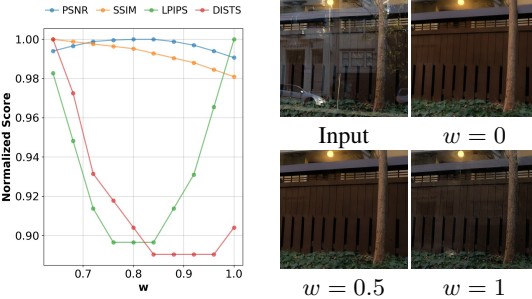

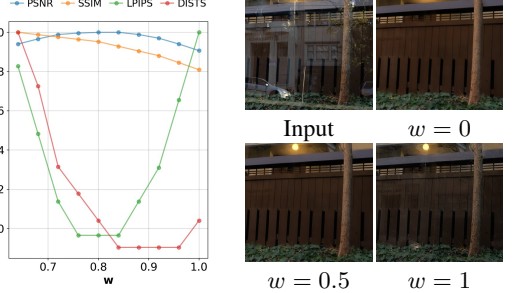

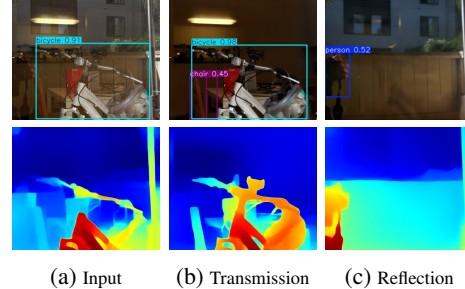

(a) Input          (b) Transmission          (c) Reflection

Figure 10: **Sensitivity analysis of the fidelity-guided feature modulation parameter** $w$. *Left*: Impact of $w$ varies across metrics. *Right*: Visual results. We select $w = 0.8$ as it provides the most balanced performance.

Figure 11: **Example of downstream tasks benefiting from reflection separation.** Our method provides high-quality separated transmission/reflection layers, improving object detection (*top*) and depth estimation (*bottom*).

in Table 3 further confirm the effectiveness of disjoint sampling, demonstrating clear performance gains.

**FGFM Parameter.** We conducted ablation experiments to investigate the effect of the FGFM parameter $w$. As shown in the left panel of Figure 10, the horizontal axis corresponds to the values of $w$, while the vertical axis represents normalized metric scores. The figure reveals that $w$ influences different metrics in distinct ways, allowing users to select $w$ according to their preferred evaluation criteria. The right panel of Figure 10 illustrates the visual impact of varying $w$: larger values yield more details and improved fidelity but may also introduce residual reflections, since FGFM does not exclusively isolate transmission information. Although reducing $w$ alleviates this issue, performance deteriorates substantially when $w < 0.5$. Considering both quantitative and qualitative aspects, we set $w = 0.8$ to achieve a balanced and overall superior performance.

**Applications.** With improved transmission and reflection estimation, our method benefits downstream tasks such as object detection and depth estimation, as demonstrated in Figure 11.

## 5 CONCLUSION

We present the first diffusion model fine-tuned for single-image reflection separation. Our unified generative diffusion framework produces accurate transmission and reflection layers using diffusion priors. We introduce a cross-layer self-attention mechanism to enhance inter-layer disentanglement, a disjoint sampling strategy to reduce residual overlaps, and latent optimization guided by a learned composition module to improve separation quality efficiently. Extensive experiments confirm superior quantitative and qualitative performance on multiple real-world datasets. Future work includes extending this method to video reflection separation and further optimizing latent-space techniques.

**Limitations.** The optimization stage requires careful parameter tuning. Occasionally, the FGFM module may fail to fully suppress reflections, introducing minor artifacts in the transmission layer.

ETHICS STATEMENT

This work does not involve human subjects, personally identifiable information, or sensitive data. All datasets used are publicly available and have been released under licenses that permit academic use. We have taken care to ensure that the proposed methods do not raise foreseeable ethical risks beyond those typically associated with research in computer vision and machine learning.

REPRODUCIBILITY STATEMENT

We provide detailed descriptions of our model architecture, training procedure, and evaluation protocols in the main text and appendix. Hyperparameters, dataset splits, and preprocessing steps are specified to enable replication of our results. Code and scripts for training and evaluation will be released with the camera-ready version to further facilitate reproducibility.

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

## A  OVERVIEW

In this supplementary material, we provide additional results to complement the main manuscript. We begin with supplementary details on our method, including the algorithmic procedure of our inference framework in appendix B and detailed formulation and discussion of the loss functions used during training in appendix C. Next, we present quantitative results for reflection and transmission layer evaluations across all datasets, using official pre-trained weights in Table 6 and Table 7. We then conduct extensive ablation studies across all datasets in appendix D. Additionally, we present an insightful experiment further illustrating the robustness and versatility of latent optimization in appendix E. Finally, we showcase visual examples demonstrating our method's effectiveness in various scenarios in appendix F.

## B  INFERENCE

Algorithm 1 details the inference procedure, including disjoint sampling and latent optimization.

---

**Algorithm 1** Reflection Disjoint Sampling Strategy

---

**Require:** Noise prediction network $\epsilon_\theta$, Encoder $\mathcal{E}$, Composition model $\mathcal{C}$
1: $z_i^{\mathcal{T}}, z_i^{\mathcal{R}} \sim \mathcal{N}(0,1), \quad z^{\mathcal{I}} \leftarrow \mathcal{E}(\mathcal{I})$
2: **for** $t = N, \dots, 1$ **do**
3: $\quad \epsilon^{\mathcal{T}} \leftarrow \epsilon_\theta(z^{\mathcal{I}}, z_i^{\mathcal{T}}, t, c^{\mathcal{T}}), \quad \epsilon^{\mathcal{R}} \leftarrow \epsilon_\theta(z^{\mathcal{I}}, z_i^{\mathcal{R}}, t, c^{\mathcal{R}})$
4: $\quad$ **if** $t \mod 5$ **then**
5: $\quad\quad$ **for** $k = 1$ to $4$ **do**
6: $\quad\quad\quad \hat{z}_0^{\mathcal{T}} \leftarrow \frac{1}{\sqrt{\bar{\alpha}_i}}\left(z_i^{\mathcal{T}} + (1 - \bar{\alpha}_i)\epsilon^{\mathcal{T}}\right)$
7: $\quad\quad\quad \hat{z}_0^{\mathcal{R}} \leftarrow \frac{1}{\sqrt{\bar{\alpha}_i}}\left(z_i^{\mathcal{R}} + (1 - \bar{\alpha}_i)\epsilon^{\mathcal{R}}\right)$
8: $\quad\quad\quad L \leftarrow \| z^{\mathcal{I}} - \mathcal{C}(\hat{z}_0^T, \hat{z}_0^{\mathcal{R}})\|_2^2$
9: $\quad\quad\quad z_i^{\mathcal{T}} \leftarrow z_i^{\mathcal{T}} - \gamma_i \|z_i^{\mathcal{T}}\| \nabla_{z_i^{\mathcal{T}}} L, \quad z_i^{\mathcal{R}} \leftarrow z_i^{\mathcal{R}} - \gamma_i \|z_i^{\mathcal{R}}\| \nabla_{z_i^{\mathcal{R}}} L$
10: $\quad\quad$ **end for**
11: $\quad$ **end if**
12: $\quad \hat{\epsilon}^{\mathcal{T}} \leftarrow \epsilon^{\mathcal{T}} + w\left(\epsilon^{\mathcal{T}} - \epsilon^{\mathcal{R}}\right), \quad \hat{\epsilon}^{\mathcal{R}} \leftarrow \epsilon^{\mathcal{R}} + w\left(\epsilon^{\mathcal{R}} - \epsilon^{\mathcal{T}}\right)$
13: $\quad z_{t-1}^{\mathcal{T}} \leftarrow \frac{1}{\sqrt{\bar{\alpha}_i}}\left(z_i^{\mathcal{T}} - \frac{1-\alpha_i}{\sqrt{1-\bar{\alpha}_i}}\hat{\epsilon}^{\mathcal{T}}\right)$
14: $\quad z_{t-1}^{\mathcal{R}} \leftarrow \frac{1}{\sqrt{\bar{\alpha}_i}}\left(z_i^{\mathcal{R}} - \frac{1-\alpha_i}{\sqrt{1-\bar{\alpha}_i}}\hat{\epsilon}^{\mathcal{R}}\right)$
15: **end for**

---

## C  LOSS FUNCTIONS

This section outlines the diffusion loss functions in our two-stage training framework. In the first stage, a combined loss for reflection and transmission layers serves as the optimization objective for the diffusion U-Net.

The diffusion model learns to invert the noising process using a noise-prediction network $\epsilon\theta(\cdot)$ by minimizing the following objective:

$$\mathbb{E}_{\epsilon \sim \mathcal{N}(0,1),t,\mathcal{I},\mathcal{T},\mathcal{R}}\left[\left\|\epsilon_t^{\mathcal{T}} - \epsilon_\theta\left(z_t^{\mathcal{T}}, z^{\mathcal{I}}, t, c^{\mathcal{T}}\right)\right\|_2^2 + \left\|\epsilon_t^{\mathcal{R}} - \epsilon_\theta\left(z_t^{\mathcal{R}}, z^{\mathcal{I}}, t, c^{\mathcal{R}}\right)\right\|_2^2\right],$$

where $c$ denotes the language prompt specifying the target layer, and $\epsilon_t^{\mathcal{T}}$ and $\epsilon_t^{\mathcal{R}}$ are sampled independently. For real training data, where ground truth for the reflection layer is unavailable, the loss is computed only for the transmission layer.

In the second stage, we optimize the Fidelity-Guided Feature Modulation (FGFM) module using a combination of L2 and LPIPS losses, formulated as:

$$\mathcal{L} = \beta_1 \left\|\hat{\mathcal{T}} - \mathcal{T}\right\|_2^2 + \beta_2 \sum_i w_i \left\|\phi_i(\hat{\mathcal{T}}) - \phi_i(\mathcal{T})\right\|_2^2, \tag{1}$$

where $\mathcal{T}$ denotes the decoded transmission, and $\hat{\mathcal{T}}$ represents the ground truth transmission. We set $\beta_1 = 1$ and $\beta_2 = 0.1$ based on empirical results.

Table 6: **Quantitative comparison of the transmission layer across different real-world datasets (Real20 (Zhang et al., 2018), Nature (Li et al., 2020), SIR$^2$ (Wan et al., 2017) using official pre-trained weights.**

| Dataset (size) | Metric | Method | | | | | | | |
|---|---|---|---|---|---|---|---|---|---|
| | | YTMT | RobustSIRR | DSRNet | RRW | DSIT | RDNet | ControlNet | Ours |
| Real20 (20) | ↑PSNR | 23.19 | 22.91 | 23.48 | 21.69 | 24.69 | 25.14 | 18.78 | **25.42** |
| | ↑SSIM | 0.800 | 0.785 | 0.788 | 0.764 | 0.813 | 0.827 | 0.645 | **0.850** |
| | ↓LPIPS | 0.131 | 0.152 | 0.136 | 0.168 | 0.118 | 0.105 | 0.282 | **0.080** |
| | ↓DISTS | 0.124 | 0.140 | 0.123 | 0.142 | 0.113 | 0.103 | 0.216 | **0.088** |
| Nature (20) | ↑PSNR | 20.90 | 21.09 | 25.01 | 25.92 | 26.36 | 25.96 | 20.04 | **26.82** |
| | ↑SSIM | 0.772 | 0.759 | 0.818 | 0.829 | 0.829 | 0.828 | 0.721 | **0.837** |
| | ↓LPIPS | 0.132 | 0.180 | 0.107 | 0.086 | 0.112 | 0.091 | 0.209 | **0.052** |
| | ↓DISTS | 0.116 | 0.147 | 0.097 | 0.085 | 0.099 | 0.085 | 0.168 | **0.066** |
| SIR$^2$ (454) | ↑PSNR | 23.98 | 22.56 | 25.86 | 25.57 | **26.62** | 26.61 | 20.77 | 25.62 |
| | ↑SSIM | 0.889 | 0.861 | 0.911 | 0.900 | **0.916** | 0.915 | 0.812 | 0.911 |
| | ↓LPIPS | 0.083 | 0.129 | 0.067 | 0.088 | 0.067 | 0.061 | 0.144 | **0.049** |
| | ↓DISTS | 0.087 | 0.120 | 0.078 | 0.087 | 0.077 | 0.072 | 0.138 | **0.067** |

Table 8: **Ablation study on the Real20 dataset.**

| C | O | D | PSNR ↑ | SSIM ↑ | LPIPS ↓ | DISTS ↓ |
|---|---|---|---|---|---|---|
| | | | 24.41 | 0.766 | 0.150 | 0.144 |
| ✓ | | | 24.28 | 0.771 | 0.139 | 0.139 |
| ✓ | ✓ | | 24.46 | 0.777 | 0.136 | 0.135 |
| ✓ | ✓ | ✓ | **25.42** | **0.850** | **0.080** | **0.088** |

Table 9: **Ablation study on the Nature dataset.**

| C | O | D | PSNR ↑ | SSIM ↑ | LPIPS ↓ | DISTS ↓ |
|---|---|---|---|---|---|---|
| | | | 25.80 | 0.769 | 0.132 | 0.129 |
| ✓ | | | 25.37 | 0.792 | 0.101 | 0.111 |
| ✓ | ✓ | | 25.80 | 0.799 | 0.101 | 0.109 |
| ✓ | ✓ | ✓ | **26.82** | **0.837** | **0.052** | **0.066** |

# D ABLATION STUDY

In the paper, we evaluate the contribution of each module: Cross-layer Self-Attention (C), Latent Optimization (O), and Disjoint Sampling (D) on the SIR$^2$ dataset. This section extends the ablation studies to all other datasets, including Real20 (Table 8) and the Nature (Table 9) dataset. The results consistently demonstrate that each component progressively enhances performance, with the full model (C+O+D) achieving the best quantitative metrics.

Table 7: **Quantitative comparison of the reflection layer on SIR$^2$ (Wan et al., 2017) using official pre-trained weights.**

| Metric | YTMT | DSRNet | DSIT | RDNet | Ours |
|---|---|---|---|---|---|
| ↑PSNR | 16.10 | 17.65 | 18.34 | 17.43 | **21.12** |
| ↑SSIM | 0.114 | 0.344 | 0.461 | 0.302 | **0.681** |
| ↓LPIPS | 0.417 | 0.370 | 0.333 | 0.373 | **0.248** |
| ↓DISTS | 0.657 | 0.445 | 0.364 | 0.351 | **0.276** |

# E AN EXPERIMENT WITH LATENT OPTIMIZATION

We provide an illustrative example to further validate the effectiveness of latent optimization guided by the proposed composition model. Specifically, we apply a text-to-image model that has not been fine-tuned for layer separation, integrating it with our proposed latent optimization. As shown in Figure 15, even with an empty prompt, the model achieves reasonable results solely under the guidance of the composition model. This result underscores the potential of composition-guided separation.

# F VISUAL RESULTS

In this section, we present additional visual results ( Figure 12, Figure 13, Figure 14). Compared to other methods, our approach produces cleaner and more accurate transmission (background) layers, effectively removing reflection artifacts while simultaneously recovering clearer and more meaningful reflection details. These results highlight significant improvements in challenging real-world scenarios.

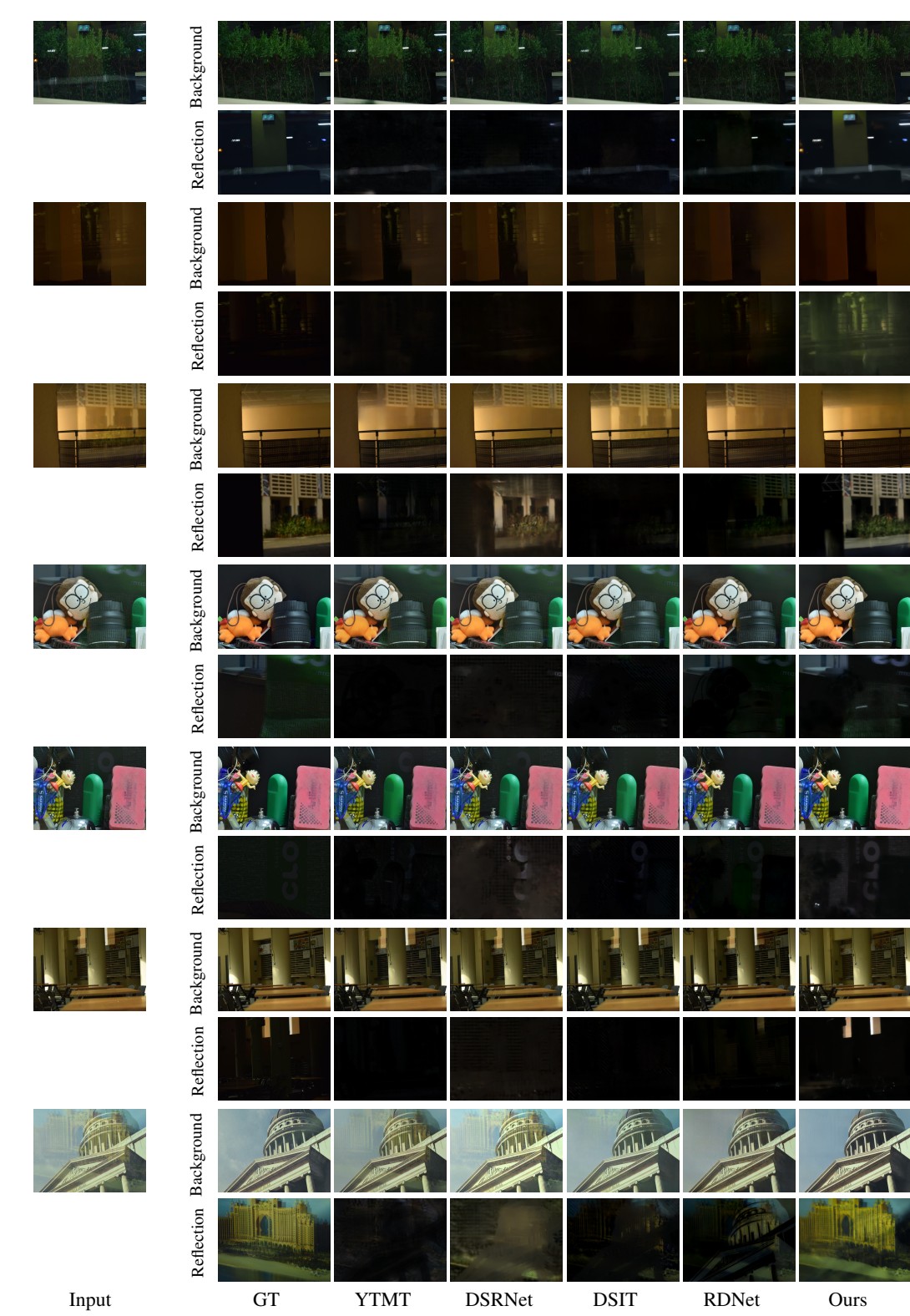

Figure 12: **Qualitative comparison of our method with state-of-the-art approaches on background-reflection separation using real-world images.**

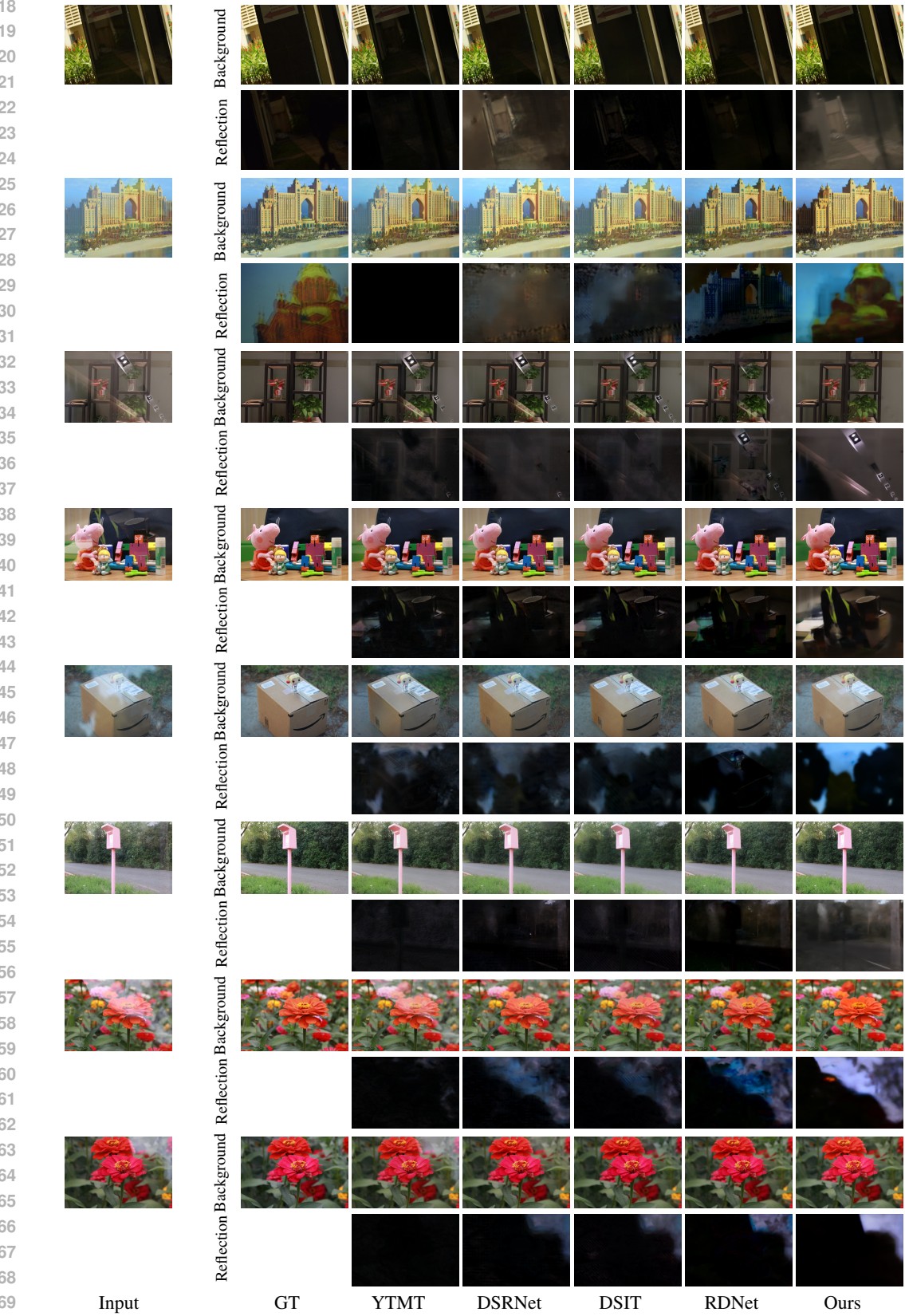

Figure 13: **Qualitative comparison of our method with state-of-the-art approaches on background-reflection separation using real-world images.**

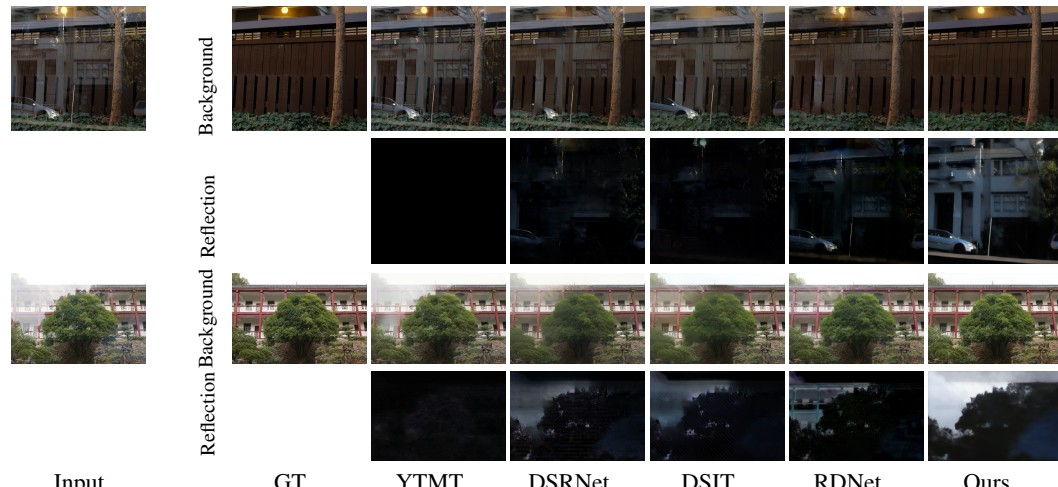

Input    GT    YTMT   DSRNet   DSIT   RDNet   Ours

Figure 14: **Qualitative comparison of our method with state-of-the-art approaches on background-reflection separation using real-world images.**

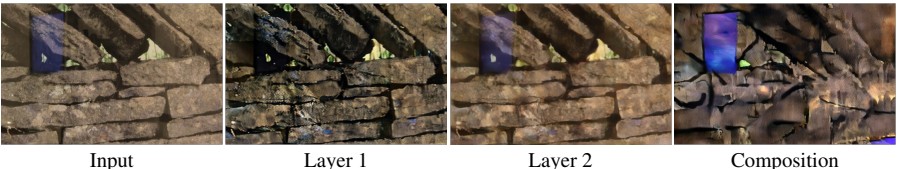

Input     Layer 1     Layer 2    Composition

Figure 15: **Layer separation by composition.**

## G  IMPLEMENTATION DETAILS.

Our training datasets consist of both synthetic and real-world images. We use the dataset from Setting 2 in Hu & Guo (2023b) and apply the same data pre-processing methods. The training process is divided into two stages. In the first stage, we fully fine-tune the diffusion model initialized from Stable Diffusion v2 (Rombach et al., 2022), using the Adam optimizer (Kingma & Ba, 2014). In the second stage, we train FGFM on output latents from the fine-tuned diffusion U-Net. To accelerate training, we limit this stage to 10 diffusion steps and apply FGFM modulation exclusively to the transmission layer since reflection layers typically contain limited high-frequency input information. For training the composition network, we utilize only synthetic data. The learning rate for all training is set to $3 \times 10^{-5}$, and the models are trained with an effective batch size of 32 on a single NVIDIA GeForce A6000 GPU. During inference, we generate results using a 50-step diffusion process.

To match the ground truth resolution, we downsample our results using area interpolation. In the official implementations of the compared methods, the ground-truth images are adjusted to the corresponding output size. To ensure a reasonable evaluation and a fair comparison, we upsample their outputs to the ground-truth resolution using bicubic interpolation when sizes do not match.

## H  IMPORTANCE OF PRE-TRAINED WEIGHTS.

We conduct an experiment to investigate the importance of pre-trained weights. For ease of comparison, we report results only on the SIR$^2$ dataset, without latent optimization or disjoint sampling. As shown in table 10, not using pre-trained weights results in significantly degraded performance. We attribute this to two main reasons. First, without pre-trained weights, the model con-

Table 10: **Effect of pre-trained weights on transmission and reflection.**

|  | PSNR ↑ | SSIM ↑ | LPIPS ↓ | DISTS ↓ |
|---|---|---|---|---|
| Train from scratch (T) | 21.52 | 0.763 | 0.154 | 0.172 |
| w/ pretrained weights (T) | **24.93** | **0.858** | **0.083** | **0.096** |
| Train from scratch (R) | 19.34 | 0.595 | 0.413 | 0.365 |
| w/ pretrained weights (R) | **20.87** | **0.659** | **0.252** | **0.285** |

verges substantially more slowly. Second, because the reflection layer lacks supervision in real-world data and its signal is typically weak in the mixed images, the absence of a generative prior hampers the model's ability to reconstruct a high-quality reflection layer.

## I    THE USE OF LARGE LANGUAGE MODELS (LLMs)

Large language models (LLMs) were employed to support the preparation of this paper and parts of the implementation. Specifically, we used LLMs for rephrasing text, generating LaTeX code snippets, and assisting with writing tasks. In addition, LLMs were used as programming aids, for example to generate and debug code templates. We reviewed, verified, and validated the final content and all experiments, and we take full responsibility for the correctness of this paper and its accompanying code.

