# OpenReview forum: "A Generative Diffusion Framework for Single Image Reflection Separation"
_ICLR.cc/2026/Conference — ICLR 2026 Conference Withdrawn Submission_

### Official Review · Reviewer_fATx · 2025-10-17

**Soundness:** 3
**Presentation:** 2
**Contribution:** 2
**Rating:** 4
**Confidence:** 4

**Summary:**

This paper proposes a diffusion-based method for reflection-transmission separation.
The core contributions appear to be:
(1) a cross-layer self-attention mechanism is introduced for feature disentanglement,
(2) a disjoint sampling strategy to reduce interference between reflection and transmission layers,
(3) and a latent optimization step with a learned composition function for improving generalization and efficiency.
The method is evaluated on three standard reflection removal datasets and demonstrates impressive results.

**Strengths:**

**Clarity and Presentation**: The paper is generally well-shaped and easy to follow, which is appreciated.

**Technical Soundness**: The core idea of leveraging a generative diffusion model for this ill-posed layer separation task is reasonable. The proposed latent-space optimization and disjoint sampling are interesting and proven to be effective.

**Impressive Qualitative Results**: Figures 6, 12, 13 provide compelling visual results, showing advantages over previous non-diffusion-based methods.

**Downstream Application Validation**: Using object detection and depth estimation tasks to validate the utility of reflection removal task demonstrates its practical value.

**Weaknesses:**

**==> Major Concerns**

I have several major concerns regarding the motivation, novelty, and experimental rigor.

**1. Motivation and Positioning:**

The paper positions itself as the first diffusion-model-based reflection separation method, which is inaccurate. There are existing diffusion-based reflection removal methods, such as (Rosh et al., 2023), and simultaneously obtaining the reflection layer is a minor differentiation. Moreover, the paper lacks a deep discussion on why a diffusion model is uniquely suited for this task, compared to other generative priors or established architectures. The introduction (and Figure 2) focuses on the limitations of existing methods, but does not provide a compelling narrative for the diffusion prior in this context. A convincing motivation should articulate the specific properties of diffusion models that are particularly effective for this specific task.


**2. Unclear Novelty and Justification of Proposed Techniques:**
The technical contributions require more accurate definitions and ablation to demonstrate their necessity and novelty.

***(1) Cross-Layer Self-Attention:*** This mechanism is claimed as a key contribution, but its formulation as a concatenation of keys/values from both layers is a relatively straightforward extension of standard self-attention. The paper does not provide a clear explanation for why this specific design is optimal for feature disentanglement in this context. The ablation results in Tables 3, 4, and the appendix (Tables 8, 9) show that this mechanism provides marginal or even negative effects on performance (e.g., PNSR). This suggests that the contribution of this mechanism is not well-understood, or may not be primary to the proposed method.

***(2) Fidelity-Guided Feature Modulation (FGFM):*** This module is proposed to address the "undesirable color drifts", but this problem is not explained and demonstrated in the main text. The novelty of the FGFM design is not explained against other feature modulation techniques. In addition, its impact is not evaluated in the primary ablation study (Table 3).

***(3) Latent Composition and Optimization:*** The latent composition function is a learned convolutional network, and the test-time latent optimization is an application of a loss function. The novelty lies in the application within the latent space for this specific task. However, the claimed improved generalization and reduced computational overheads require stronger supporting evidence.


**3. Insufficient and Potentially Unfair Evaluation:**
The experimental setup and analysis require significant strengthening to support the paper's claims.

***(1) Benchmarks and Baselines:***
The choice of evaluation datasets is limited. Omitting the recent large-scale benchmark (e.g., Zhu et al., 2024) weakens the claim of state-of-the-art performance. The selection of competing methods is also questionable. Including RobustSIRR (focused on adversarial robustness) is not a strong choice for pure performance comparison. Most critically, as a diffusion-based method, it is essential to compare against a wider range of recent diffusion-based techniques from related image enhancement tasks (e.g., shadow, haze, or rain removal) adapted for this problem. Comparing primarily against older, non-generative methods does not convincingly demonstrate the advantage of the proposed diffusion-based method.

***(2) Marginal Gains and Baseline Strength:***
Table 1 shows that the performance gains over non-diffusion methods are marginal. Moreover, the "w/o C, O, D" baseline in Table 3 (which is essentially a fine-tuned Stable Diffusion v2) already outperforms five of the competing methods. This raises a critical question: Are the reported gains due to the novel contributions, or simply the raw power of the underlying foundational model? A more rigorous baseline would be a straightforward fine-tuning of Stable Diffusion v2 (or more powerful models such as Stable Diffusion 3.5 and DiTs) without the proposed modules. The comparisons feel unfair, picking a modern foundation model against older architectures.

***(3) Incomplete Ablations and Analysis:***
First, the effect of the FGFM module is not studied in Table 3. Second, the claim about computational efficiency is supported by a single internal comparison (latent vs. pixel optimization). To claim general efficiency, a comparison of FLOPs/inference time/GPU consumption between ablated models and against other methods is needed. Third, Figure 10 contradicts the claim that residual reflections are controlled by w, as they appear (see top-right regions) even when w=0. This needs explanation.

***(4) Superficial Limitations Section:***
The current discussion on limitations is brief and attributes failures only to a hyperparameter. A more honest reflection would include specific failure cases (e.g., certain types of reflections or textures it cannot handle) and discuss the intrinsic limitations of the approach, such as its performance on truly "in-the-wild" images not represented by the training data.


**==> Typos and Minor Suggestions**

***(1) Figure Order:*** Please check the figure references, as Figure 2 is cited before Figure 1, and Figure 9 before Figure 8.

***(2) Term Use:*** "Low-light reflection" (Line 79) should be rephrased to "reflections in low-light scenes" or similar.

***(3) Naming Consistency:*** The naming of the core contributions is inconsistent between the abstract, introduction, and contribution sections. Please use a single, clear set of terms throughout (e.g., "cross-layer self-attention," "disjoint sampling," and "latent optimization").

***(4) References Redundancy:*** There are redundant references (e.g., Hu & Goo, 2023a/b). Please consolidate.

**==> Justification**

This paper presents an interesting application of diffusion models to reflection removal with promising qualitative results. However, the paper needs to rigorously establish its novelty over simple fine-tuning of a foundation model, provide a more comprehensive and fair experimental comparison against modern baselines (especially other diffusion-based methods), and thoroughly justify the design and necessity of each proposed component through clear ablations and analysis. The fundamental issue is to disentangle the contribution of the powerful baseline model from the novel architectural components.

Given that the concerns now outweigh the strengths, I recommend a weak rejection at this point.

**Questions:**

1. How does the proposed cross-layer self-attention lead to better feature disentanglement than, for instance, two separate self-attentions? Given its mixed results in the ablations, what is the definitive evidence of its necessity?

2. Can authors provide an ablation study for the FGFM module in the main results table (Table 3) to quantify its impact on both quantitative metrics and qualitative color fidelity?

3. The "w/o C, O, D" baseline is very strong. Could you include a more direct baseline of simply fine-tuning Stable Diffusion v3.5 (or DiT) for this task and show that the proposed method still has an advantage over it?

4. Given the rapid progress in foundational models, have the authors experimented with more powerful backbones like Stable Diffusion 3.5 or DiTs? What the proposed techniques should be seen as, a set of plugins that are largely dependent on the base model's capability or that can continuously improve the performance?

---

### Official Review · Reviewer_isHZ · 2025-10-31

**Soundness:** 2
**Presentation:** 3
**Contribution:** 2
**Rating:** 4
**Confidence:** 4

**Summary:**

This paper introduces a generative framework for single-image reflection separation (SIRS), leveraging a fine-tuned latent diffusion model. The core idea is to simultaneously generate the transmission and reflection layers from a single input image. The proposed method incorporates several novel components: a cross-layer self-attention mechanism to improve feature disentanglement, a disjoint sampling strategy to enforce mutual exclusivity between the two layers, and a test-time latent optimization step guided by a learned composition function to enhance fidelity. The authors present experiments on multiple real-world datasets, claiming that their approach achieves superior performance over existing state-of-the-art methods in both quantitative metrics and perceptual quality.

**Strengths:**

1.  The proposed strategy of using a diffusion model to predict two mutually exclusive components within a single image is novel and interesting.

2.   The paper is well-structured and clearly written.

3.   The visual results demonstrate that the method reconstructs reflection layers that are clearer and more complete than those from previous methods.

**Weaknesses:**

1.  The rationale behind the disjoint sampling strategy, while sounding plausible, rests on the assumption of semantic-level mutual exclusivity between the transmission and reflection layers. However, reflections in real-world scenarios are frequently sparse or non-semantic artifacts, such as simple light glares or blurs, which lack distinct semantic content that can be semantically opposed to the transmission layer. It is doubtful whether a semantic-level exclusion can be reliably enforced for these common reflection types without additional guidance, such as natural language descriptions.


2.  Related to the above, the reliance on fixed, single-word prompts ("Transmission" and "Reflection") is a significant concern. It is highly questionable whether the pre-trained Stable Diffusion 2 model has a robust and correctly aligned understanding of the concept "Transmission" in this context. While the model may have some notion of "Reflection," the concept of the underlying, intended scene ("Transmission") is abstract and likely not well-represented in its training data. A more plausible approach would involve using rich, descriptive text for each layer, but this is not explored and links back to the ill-posed nature of describing non-semantic reflections.


3.  The method's ability to model the reflection layer appears heavily dependent on the availability of ground-truth reflection images. This is a major limitation, as such data is scarce in existing real-world datasets and expensive to acquire. The authors acknowledge in the supplementary material that for real training data, the loss is only computed on the transmission layer. This significantly undermines confidence in the model's ability to accurately model the reflection layer for real-world images. Furthermore, the claim of generalization is not well-supported. The out-of-training-domain examples in Figure 1 do not seem to represent the claimed "challenging scenarios like strong reflections or complex overlapping content" and are presented without any comparison to existing SOTA methods. This makes it impossible to verify the method's performance on challenging, in-the-wild images.


4.  Despite claims of superiority, the method's overall quantitative performance is not consistently state-of-the-art. A weighted average of the results from Table 1 across real20 and SIR^2 datasets shows that the proposed method (PSNR ≈ 25.61, SSIM ≈ 0.910) is outperformed by both DSIT (PSNR ≈ 26.49, SSIM ≈ 0.922) and RDNet (PSNR ≈ 26.65, SSIM ≈ 0.917). This discrepancy suggests the existence of significant failure cases that negatively impact the overall average. This is particularly concerning given the inclusion of the FGFM module, which is specifically designed to maintain fidelity. The authors should provide an explanation for this performance gap.


5.  The ablation study presented in Table 4 for the cross-layer self-attention mechanism, a key architectural contribution, fails to convincingly demonstrate its importance. The inclusion of this module results in a negligible PSNR improvement of ~0.3% (20.81 to 20.87) while causing a significant degradation in SSIM of ~4% (0.688 to 0.659). This result appears to contradict the claim that this mechanism is effective and raises questions about its design.


6.  The paper suffers from a lack of crucial implementation details, which hinders reproducibility and a full assessment of its technical soundness. For example, the specific architecture of the composition network is not described, the exact composition of the training dataset is not clearly specified, and a justification for not comparing against the highly relevant diffusion-based method, L-DiffER, is absent. These omissions reduce the technical validity of the work.

**Questions:**

1.  Could you elaborate on how the disjoint sampling strategy handles non-semantic reflections (e.g., light glares, blurs)? How can you be certain that the fixed prompts "Transmission" and "Reflection" provide meaningful and distinct guidance to the SD2 model for this task?

2.  Given that the reflection loss is not utilized for real training data, what ensures that the model learns to generate accurate reflections for in-the-wild images? Could you provide direct comparisons against SOTA methods on the challenging real-world cases shown in Figure 1 to substantiate your generalization claims?

3.  How do you explain the lower weighted average PSNR and SSIM scores in Table 1 compared to methods like DSIT and RDNet? Are there specific types of scenes or reflections where your method underperforms, leading to this lower average?

4.  The results in Table 4 indicate that the cross-layer self-attention module degrades SSIM performance. Could you please provide a justification for this design choice in light of these results? Does it perhaps offer a qualitative benefit that is not captured by these metrics?

5.  Could you please provide the architectural details of the composition network, specify the training data composition and sources, and explain why a comparison with the L-DiffER method was omitted from your evaluation?

---

### Official Review · Reviewer_6rRb · 2025-11-01

**Soundness:** 2
**Presentation:** 3
**Contribution:** 2
**Rating:** 4
**Confidence:** 4

**Summary:**

This paper presents a generative framework based on diffusion models for the task of single-image reflection separation. The authors introduce several key components: 1) a cross-layer self-attention mechanism to enhance feature disentanglement between the two layers; 2) a disjoint sampling strategy to iteratively reduce interference during the diffusion process; and 3) a test-time latent optimization step guided by a learned composition function to refine the results. The proposed method demonstrates good quantitative performance on the reported benchmarks.

**Strengths:**

1. The paper introduces a dual-stream interactive mechanism for reflection decomposition into a diffusion model, along with the design of several auxiliary modules.
2. The proposed method achieves good performance on the reported experimental results.

**Weaknesses:**

1. The specific role of the dual-stream interaction within the Diffusion Model requires a more detailed explanation and experimental validation. Can the role of this interaction be verified by visualizing intermediate features or by statistically analyzing the contributions of the interactive features to the two branches?
2. The authors need to provide specific information about the pre-trained VAE and Diffusion models used. They also need to describe the training details more thoroughly. Additionally, a question remains: does improving the quality of the reflection layer require training?
3. The paper does not compare with DAI (Dereflection Any Image with Diffusion Priors and Diversified Data). Since DAI is currently a preprint, a discussion of the differences and advantages would be sufficient.
4. The authors claim in the contributions section that their method "greatly reduces computational resources," but the paper provides no comparison of model complexity (parameters), computational load (GFLOPs), or inference time to support this claim.
5. The paper needs to show more visual results on challenging real-world cases to better validate the model's performance. The authors could compare against examples from the RDNet and DAI papers/repositories or provide results on newly captured images.
6. What is the difference between Table 1 and Table 6? Why are your retrained results lower than those reported in the original authors' papers? What are your training configs and evaluating details?
7. The performance of the proposed method does not seem to offer a significant improvement over non-generative methods. What, then, is the advantage of using a generative approach for reflection separation?
8. Will there be any hallucinations when testing the model on OOD's data?
9. Will changing different random seeds result in inconsistent results, sometimes good and sometimes bad?

**Questions:**

Please refer to Weaknesses.

---

### Official Review · Reviewer_VHVx · 2025-11-02

**Soundness:** 2
**Presentation:** 1
**Contribution:** 2
**Rating:** 2
**Confidence:** 4

**Summary:**

This paper proposes a diffusion based method to tackle single image reflection removal task. The proposed method has come up with a few modification on the existing diffusion frameworks to enhance the capability of separate transmission layer and reflection layer from an input image. It introduces cross-layer self-attention module and disjoint sample to split the latent embedding processing. In addition to the attention module, it also builds a latent composition function that tries to optimize the latent vaiable of a composite reflection and transmission with ground truth latent. Extensive experiments demonstrate the proposed method outperforms the baseline methods.

**Strengths:**

1. The quantitative experiment results show that the proposed method performs better than baseline methods consistently.

**Weaknesses:**

1. Figure 4 is not clear. What does upper and lower figure diagram refer to respectively?
2. The quantitative performance of the baseline are not aligned with the corresponding published results. Example, DSIT should have 26.49 / 0.922 (PSNR/SSIM) on SIR^2 datasets.
3. Line 245 - "This cross-layer attention structure encourages queries to leverage information from both layers, effectively disentangling the two layers." This claims requires more evidences, support and reasoning. i.e. Why does cross-layer attention disentangle transmission and reflection by querying both key values?
4. The description for disjoint sampling and separation optimization by composition in Sect 3.3 are difficult to understand.  For example, what is the usage of latent composition function? How does it connect with the other modules in the model?  Line 259, why does the equation help to separate reflection and transmission? Can you please explicit describe relationships of the variables logically?

**Questions:**

The main issue is the presentation of the method. The method section 3. lacks coherent structure and logical flow. The arguments presented do not follow a clear progression, and the connections between ideas remain underdeveloped or absent entirely. Key points appear disjointed, making it difficult for readers to follow the intended line of reasoning.

---

### Note · Authors · 2025-11-13

I have read and agree with the venue's withdrawal policy on behalf of myself and my co-authors.